 RESEARCH ARTICLE  | 

# SARS-CoV-2 NSP13 interacts with TEAD to suppress Hippo-YAP signaling

Fansen Meng[1], Jong Hwan Kim[2], Chang-Ru Tsai[3], Jeffrey D Steimle[3], Jun Wang[2], Yufeng Shi[2], Rich G Li[1], Bing Xie[3], Vaibhav Deshmukh[3], Shijie Liu[4,5], Xiao Li[1], James F Martin[1,2,3]*

[1]McGill Gene Editing Lab, The Texas Heart Institute, Houston, United States; [2]Cardiomyocyte Renewal Laboratory, The Texas Heart Institute, Houston, United States; [3]Department of Integrative Physiology, Baylor College of Medicine, Houston, United States; [4]The Heart Institute, Division of Molecular Cardiovascular Biology, Cincinnati Children's Hospital Medical Center, Cincinnati, United States; [5]Department of Pediatrics, University of Cincinnati College of Medicine, Cincinnati, United States

## eLife Assessment

This **important** study elucidates the molecular function of the SARS-CoV-2 helicase NSP13, which inhibits the transcriptional activity of the YAP/TEAD complex *in vitro* and *in vivo*. The evidence supporting the authors' claims is **compelling**, based on cell biological assays and multi-omics studies. This work contributes to the understanding of the new regulatory mechanism of YAP/TEAD after SARS-CoV-2 infection and will be of interest to researchers investigating COVID-19 infection and the Hippo-YAP signaling pathway.

*For correspondence:
jfmartin@bcm.edu

**Abstract** The Hippo pathway controls organ development, homeostasis, and regeneration primarily by modulating YAP/TEAD-mediated gene expression. Although emerging studies report Hippo-YAP dysfunction after viral infection, it is largely unknown in the context of severe acute respiratory syndrome coronavirus 2 (SARS-CoV-2). Here, we analyzed RNA sequencing data from human-induced pluripotent stem cell-derived cardiomyocytes (iPSC-CMs) and SARS-CoV-2-infected human lung samples, and observed a decrease in YAP target gene expression. In screening SARS-CoV-2 nonstructural proteins, we found that nonstructural protein 13 (NSP13), a conserved coronavirus helicase, inhibits YAP transcriptional activity independent of the upstream Hippo kinases LATS1/2. Consistently, introducing NSP13 into mouse cardiomyocytes suppresses an active form of YAP (YAP5SA) *in vivo*. Subsequent investigations on NSP13 mutants revealed that NSP13 helicase activity, including DNA binding and unwinding, is crucial for suppressing YAP transactivation in HEK293T cells. Mechanistically, TEAD4 serves as a platform to recruit NSP13 and YAP. NSP13 likely inactivates the YAP/TEAD4 transcription complex by remodeling chromatin to recruit proteins, such as transcription termination factor 2 (TTF2), to bind the YAP/TEAD/NSP13 complex. These findings reveal a novel YAP/TEAD regulatory mechanism and uncover molecular insights into Hippo-YAP regulation after SARS-CoV-2 infection in humans.

## Introduction

The evolutionarily conserved Hippo-YAP signaling pathway integrates various extracellular signals, including mechanical force, cell adhesion, and nutrient availability, through the protein kinases LATS1/2 (*Ma et al., 2019*). YAP is the primary substrate of LATS kinases, and following phosphorylation, is inactivated via cytoplasmic retention or degradation. Unphosphorylated YAP translocates to

the nucleus and complexes with transcription factor partners, notably TEAD, to activate target genes (*Ma et al., 2019*; *Wang et al., 2018*) involved in tissue development, homeostasis, and regeneration across multiple organs (*Wang et al., 2018*; *Meng et al., 2022*; *Russell and Camargo, 2022*; *Hong et al., 2016*; *Moya and Halder, 2019*; *Zhang et al., 2018*). YAP/TEAD is regulated by intricate molecular mechanisms that govern its activity, localization, and interaction. Dysregulation of these processes has been implicated in various diseases, including cancer (*Paul et al., 2022*; *Pocaterra et al., 2020*; *Lin et al., 2017*; *Huh et al., 2019*).

Recent studies reveal that YAP is an endogenous brake of innate antiviral immunity (*Wang et al., 2017*; *Zhang et al., 2017*). Genetic loss of YAP enhances antiviral responses, whereas expression of a transactivation-deficient mutant (YAP-6SA) suppresses these responses. Conversely, multiple canonical YAP target genes, such as *Il-6*, *Ccl2*, and *Csf1*, modulate innate-immune signaling and cytokine production (*Meli et al., 2023*; *Wang et al., 2020*; *Xiao et al., 2019*). Notably, different types of viral infection distinctly influence YAP expression, degradation, and nuclear localization (*Wang et al., 2019*). Although these observations shed light on the complex dynamics of YAP during viral responses, the mechanisms underlying YAP/TEAD regulation after SARS-CoV-2 infection are poorly understood. SARS-CoV-2, the center of the COVID-19 pandemic, has well-documented effects on the respiratory, digestive, central nervous, and cardiovascular systems (*Ma et al., 2020*; *De Felice et al., 2020*; *Nishiga et al., 2020*). Investigating SARS-CoV-2 and Hippo-YAP signaling provides molecular insight into virus-induced pathophysiology. Two groups recently reported opposite effects on YAP activity after SARS-CoV-2 infection (*Garcia et al., 2022*; *Pinto et al., 2023*). However, the lack of YAP/TEAD target gene expression data in these studies prevents a full understanding of these disparate results. Thus, the precise understanding of YAP/TEAD regulation after SARS-CoV-2 infection remains elusive.

Here, we found that SARS-CoV-2 infection reduces YAP target gene expression in human-induced pluripotent stem cell-derived cardiomyocytes (hiPSC-CMs) and lung epithelial cells of COVID-19 patients. By comprehensively screening SARS-CoV-2 nonstructural proteins, we identified NSP13 as a key factor in inhibiting YAP transcriptional activity and suppressing active YAP5SA activity *in vivo*. Moreover, NSP13 suppression of YAP is dependent on its helicase activity, where both DNA binding and unwinding are required. Mechanistically, NSP13 is directly bound to TEAD4 and inhibits the transcriptional activity of the YAP/TEAD4 complex by remodeling chromatin and recruiting transcriptional suppressors such as TTF2. These findings reveal a novel function of NSP13 in regulating YAP/TEAD activity and provide key insights into how the SARS-CoV-2 genome modulates transcriptional activity of the YAP-TEAD complex.

## Results

### SARS-CoV-2 infection suppresses YAP activity in host cells

To assess YAP activity following SARS-CoV-2 infection *in vitro*, we evaluated a bulk RNA-sequencing dataset derived from hiPSC-CMs exposed to varying SARS-CoV-2 concentrations (*Perez-Bermejo et al., 2021*; *Figure 1A*). Cardiomyocyte-specific YAP target gene (*Monroe et al., 2019*) expression revealed that YAP targets were decreased in a dose-dependent manner after SARS-CoV-2 infection (*Figure 1B*). YAP is known to directly regulate several innate-immunity genes in mouse, including *Ccl2*, *Thbs1*, and *Csf1* (*Wang et al., 2020*; *Xiao et al., 2019*). We observed that these genes are downregulated in SARS-CoV-2-infected cells, mirroring the decrease in canonical YAP targets such as *VGLL3*, *AMOTL2*, and *CCN1* (*Figure 1C*).

We then assessed YAP activity with SARS-CoV-2 infection *in vivo* by integrating and reanalyzing single-nuclei RNA (snRNA) sequencing data derived from human lung samples (*Delorey et al., 2021*; *Melms et al., 2021*). We identified 10 major cell types (*Figure 1D*, *Figure 1—figure supplement 1*). *TMPRSS2* and *ACE2*, two key entry factors for SARS-CoV-2 infection (*Hoffmann et al., 2020*), were more highly expressed in lung epithelial cells compared to other cell types (*Figure 1E*, *Figure 1—figure supplement 2*). Lung epithelial cells were categorized into alveolar type 1 (AT1) and alveolar type 2 (AT2) (*Figure 1—figure supplement 3*). SARS-CoV-2 is more likely to infect AT1 cells as they cover >95% of the alveolar surface. We performed an unbiased GO analysis of the differentially expressed genes (DEGs) and observed that genes associated with or regulating the Hippo pathway were downregulated in AT1 cells from COVID-19 patients (*Figure 1F*). Consistent with previous

**Figure 1.** SARS-CoV-2 infection suppresses YAP activity. (**A**) Overview of SARS-CoV-2 infection in human induced pluripotent stem cell-derived cardiomyocytes (hiPSC-CMs). (**B**) Box plot showing the mean expression scores of known YAP target genes in hiPSC-CM bulk RNA-sequencing data. Each dot represents a biological replicate. Student's *t*-test; **p<0.01, ****p<0.0001. (**C**) Heatmap displaying the expression Z-scores of example YAP targets in the iPSC-CM bulk RNA-sequencing data. Each column corresponds to a single biological sample. (**D**) Overview of integrated single-nucleus RNA sequencing and uniform manifold approximation and projection (UMAP) of cell types in lung samples from controls and patients with COVID-19. EC, endothelial cells; NK, natural killer cells; SMC, smooth muscle cells. (**E**) UMAP visualization of TMPRESS2 expression in the 10 cell types. (**F**) GO analysis of downregulated genes in AT1 cells from COVID patients. (**G**) Yap scores in alveolar type 1 (AT1) and alveolar type 2 (AT2) epithelial cells from lung samples in controls and patients with COVID-19. Wilcoxon test, ****p<0.0001. (**H**) Expression of Yap targets in AT1 and AT2 cells of control and COVID-positive lung samples.

The online version of this article includes the following figure supplement(s) for figure 1:

**Figure supplement 1.** Heatmap showing the most highly expressed markers for each cell type in human lung samples.

**Figure supplement 2.** Uniform manifold approximation and projection (UMAP) visualization showing ACE2 expression in all cell types.

**Figure supplement 3.** Epithelial cell subtypes AT1 and AT2 with marker genes.

**Figure supplement 4.** Expression of Yap targets genes involved in innate immune response or regulation in AT1 and AT2 cells from control and COVID-positive lung samples.

data *in vitro*, the YAP score, evaluated via YAP target gene expression, was lower in AT1 cells from COVID-19 patients than in controls (*Figure 1G*). Signature genes associated with AT1 cells, including the reported YAP target genes *AGER* and *CLIC5* (*Penkala et al., 2021*; *Gokey et al., 2021*), were downregulated in COVID-19 patient lung samples (*Figure 1H*). These data support the notion that SARS-CoV-2 infection suppresses YAP activity in host cells. However, further analysis revealed that

multiple YAP target genes involved in innate immunity and cytokine signaling were paradoxically elevated (*Figure 1—figure supplement 4*). This discrepancy likely reflects a combination of factors, including cell-type specificity, activation of parallel signaling pathways, and alterations in mechanical cues or tissue architecture that can independently drive expression of these genes.

## NSP13 inhibits YAP5SA transactivation

To identify which SARS-CoV-2 protein suppresses YAP activity, we screened 11 NSP proteins using a dual-luciferase reporter assay. This assay measures firefly and Renilla luciferase activities sequentially from the same sample. Normalization to Renilla luciferase serves as an internal control, ensuring accuracy and reproducibility of the results. Compared with other NSPs, NSP13 strongly inhibited YAP transcription activity (*Figure 2A*). By using a constitutively active form of YAP (YAP5SA) that is resistant to phosphorylation and inactivation by LATS1/2 (*Zhao et al., 2007*), we observed that NSP13 suppresses YAP transactivation independent of its upstream kinase LATS2 (*Figure 2—figure supplement 1*). Moreover, NSP13 attenuated YAP5SA activity dose-dependently (*Figure 2B*). A recent study reported that NSP13 suppresses episomal DNA transcription, as evidenced by reduced Renilla luciferase activity and decreased GFP expression upon co-expression with NSP13 (*Li et al., 2023*). To evaluate this in our system, we examined our raw Renilla luciferase data and found that while 100 ng of NSP13 had no effect, 400 ng of NSP13 reduced Renilla luciferase levels by approximately 50% compared to the YAP5SA-only group (*Figure 2—figure supplement 2*). Notably, the firefly luciferase signal, driven by YAP/TEAD interaction (HOP-Flash), exhibited an even greater reduction. To assess the specificity of this suppression, we also conducted a Notch reporter assay and observed that co-expression of NSP13 with NICD (Notch intracellular domain) did not inhibit Notch signaling (*Figure 2—figure supplement 3*). These results reveal a specific suppressive effect of NSP13 on YAP-mediated transcriptional activity.

To investigate NSP13 function *in vivo*, we used YAP5SA transgenic mice (aMyHC-MerCreMer;YAP5SA), which express YAP5SA in cardiomyocytes following tamoxifen administration. YAP5SA induction produces cardiomyocyte hyperplasia and increased ejection fraction (EF) and fractional shortening (FS), ultimately leading to mortality (*Monroe et al., 2019*). We used the adenovirus expressing NSP13 (AAV9-NSP13), which specifically infects cardiomyocytes (*Figure 2—figure supplement 4*, *Figure 2C*). NSP13-induced expression in YAP5SA mouse cardiomyocytes increased survival rates (*Figure 2D*) and restored cardiac function (*Figure 2E*, *Figure 2—figure supplement 5*). Furthermore, NSP13 expression reversed the smaller left ventricle (LV) chamber in YAP5SA mouse hearts (*Monroe et al., 2019*; *Figure 2F and G*). To investigate NSP13 induction in the mouse heart further, we collected heart tissue from surviving mice 21 days after tamoxifen injection. Notably, NSP13 expression in YAP5SA mice reversed heart overgrowth (*Figure 2H and I*, *Figure 2—figure supplement 6*). Together, these data indicate that NSP13 suppresses YAP activity *in vitro* and *in vivo*.

## NSP13 helicase activity is required for YAP suppression

NSP13, a helicase with conserved sequence across all coronaviruses (*Figure 3A*), plays a critical role in viral replication and is a promising target for antiviral treatment (*Yan et al., 2021*; *Chen et al., 2020*; *Yan et al., 2020*; *Newman et al., 2021*; *Chen et al., 2022*; *Malone et al., 2021*; *Nandi et al., 2022*; *Zeng et al., 2021*). K131, K345/K347, and R567 were identified as important amino acid sites for NSP13 helicase activity in SARS-CoV (*Jia et al., 2019*). Given the 99.8% sequence identity of NSP13 between SARS-CoV-2 and SARS-CoV, we speculated that these sites were similarly crucial for SARS-CoV-2 NSP13. Reporter assays revealed that NSP13-K131A, which retains partial helicase activity, can suppress YAP, whereas NSP13-R567A (no ATP consumption) and NSP13-K345A/K347A (obstructed nucleic acid binding channel) failed to inhibit YAP transcriptional activity (*Figure 3B*). To assess the contribution of each domain to YAP regulation, we constructed six NSP13 truncations (*Figure 3C*), yet none of these truncations reduced YAP transactivation (*Figure 3D*). These findings suggest that full-length NSP13 with helicase activity (DNA binding and ATP-dependent unwinding) is required to suppress YAP transactivation, and that partial helicase function is sufficient to fully inhibit YAP.

SARS-CoV-2 has consistently mutated over time, resulting in variants that differ from the original virus. We evaluated NSP13 mutations (*Newman et al., 2021*) and observed that all mutants can suppress YAP transactivation (*Figure 3E and F*, *Figure 3—figure supplement 1*).

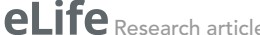

**Figure 2.** NSP13 inhibits YAP transactivation. (**A**) Screening of 11 NSPs for YAP activation by using a dual-luciferase reporter assay (HOP-flash). Compared with other NSPs, NSP13 strongly inhibited YAP transactivation at low protein expression levels. (n=3 independent experiments; data are reported as mean ± SD). ****p<0.0001, one-way ANOVA. (**B**) Reporter assay (8xGTIIC) results showing that NSP13 but not YAP upstream kinase LATS2 inhibited YAP5SA transactivation in a dose-dependent manner. (n=3 independent experiments; data are presented as the mean ± SD). ***p<0.001, ****p<0.0001, one-way ANOVA. (**C**) Experimental design of NSP13 study in mice. Control (aMyHC-MerCreMer;WT) and YAP5SA (aMyHC-MerCreMer;YAP 5SA) mice were injected with AAV9-GFP or AAV9-NSP13. At 12 days after virus injection, the mice received two low doses of TAM (10 ug/g). Cardiac function was recorded by echocardiography at days 4 and 8 after the second shot of tamoxifen. Hearts of all surviving mice were collected at day 21 post-tamoxifen injection. (**D**) NSP13 expression in cardiomyocytes improved the survival rate of YAP5SA mice after TAM injection compared to YAP5SA mice with AAV9-GFP infection. **p=0.0099, log-rank (Mantel–Cox) test. (**E**) Ejection fraction in YAP5SA mice was increased on day 8 after tamoxifen injections (10 ug/g x2). NSP13 expression reversed the increase of EF in YAP5SA mice. (n=6 in MCM + GFP, MCM + NSP13, and YAP5SA + GFP group; n=8 in YAP5SA + NSP13 group). ****p<0.0001, three-way ANOVA. (**F**) Representative B-mode and M-mode echocardiographic images of mouse hearts in four groups, 8 days after tamoxifen (TAM) induction. (**G**) A reduction in the left ventricle size was seen in YAP5SA mice at day 8 after tamoxifen injection. NSP13 introduction reversed this trend as evidenced by an increase in the diameter of the left ventricle (n=6 in MCM + GFP, MCM + NSP13, and YAP5SA + GFP group; n=8 in YAP5SA + NSP13 group). ***p<0.001, three-way ANOVA. (**H, I**) Representative whole mount and H&E images of mouse hearts at 21 days after tamoxifen induction. Scale bar, 2 mm.

The online version of this article includes the following source data and figure supplement(s) for figure 2:

**Source data 1.** Original files for western blot analysis displayed in *Figure 2A*.

**Source data 2.** Original western blots for *Figure 2A*, indicating the relevant bands and treatments.

**Source data 3.** Original files for western blot analysis displayed in *Figure 2B*.

**Source data 4.** Original western blots for *Figure 2B*, indicating the relevant bands and treatments.

*Figure 2 continued on next page*

*Figure 2 continued*

**Figure supplement 1.** Reporter assay (HOP-flash) results indicating that NSP13 inhibited YAP5SA transactivation at low protein expression levels when compared with other NSPs.

**Figure supplement 1—source data 1.** Original files for western blot analysis displayed in *Figure 2—figure supplement 1*.

**Figure supplement 1—source data 2.** Original western blots for *Figure 2—figure supplement 1*, indicating the relevant bands and treatments.

**Figure supplement 2.** Raw reads of Renilla luciferase in *Figures 2A and 3B and D*.

**Figure supplement 3.** Reporter assay (Notch reporter) results showing that NSP13 cannot suppress NICD activation.

**Figure supplement 4.** NSP13 expression at different time points in mouse hearts after AAV9-HA-NSP13 virus injection.

**Figure supplement 4—source data 1.** Original files for western blot analysis displayed in *Figure 2—figure supplement 4*.

**Figure supplement 4—source data 2.** Original western blots for *Figure 2—figure supplement 4*, indicating the relevant bands and treatments.

**Figure supplement 5.** Fractional shortening was increased in YAP5SA mice at day 8 after tamoxifen injection (10 ug/g x2).

**Figure supplement 6.** The overgrowth of the heart in YAP5SA mice, as evidenced by heart/body ratio, was reversed by NSP13 expression.

## NSP13 interacts with TEAD4 and recruits YAP repressors

To determine the molecular mechanism(s) underlying NSP13 suppression of YAP, we performed immunofluorescence (IF) analyses. Our data revealed that NSP13 localizes to the cytoplasm and nucleus, whereas most YAP5SA colocalizes with NSP13 in the nucleus three days after tamoxifen injection (*Figure 4A*). However, co-immunoprecipitation (co-IP) assays revealed no direct interaction between NSP13 and YAP5SA (*Figure 4—figure supplement 1*). Additional assays indicated that NSP13 is associated with the transcription factor TEAD4 (*Figure 4B*), and both the N- and C-terminal ends of TEAD4 interact with NSP13 (*Figure 4—figure supplement 2*), suggesting that NSP13 prevents YAP transactivation by competitively interacting with TEAD4. Unexpectedly, introduction of NSP13 had no effect on the YAP/TEAD4 association (*Figure 4—figure supplement 3*). However, the YAP/NSP13 interaction was much stronger in the presence of TEAD4, indicating that TEAD4 is a platform to recruit YAP and NSP13 (*Figure 4C*). Consistent with this, IF analysis of HeLa cells and heart sections revealed that YAP5SA is restricted to the nucleus when co-expressed with either NSP13 WT or R567A, in contrast to the vector control (*Figure 4D*, *Figure 4—figure supplement 4*). Moreover, NSP13 protein levels in cardiomyocytes accumulated following YAP5SA induction (*Figure 4D and E*), suggesting that nuclear NSP13/YAP/TEAD4 prevents NSP13 degradation. Given that NSP13 preferentially interacts with TEAD4 in the nucleus (*Figure 4B-C*, *Figure 4—figure supplement 3*), we next determined whether this association was DNA-dependent using multiple nucleases: Universal Nuclease (which degrades all forms of DNA and RNA), DNase I (which cleaves both single- and double-stranded DNA), and RNase H (which selectively cleaves the RNA strand in RNA/DNA hybrids). These treatments did not disrupt the NSP13/TEAD4 interaction (*Figure 4—figure supplement 5*), revealing that their binding is not dependent on nucleic acids.

Given that NSP13 forms a complex with YAP and TEAD4 (*Figure 4C*, *Figure 4—figure supplement 3*), we hypothesized that it recruits nuclear repressors to inhibit YAP transactivation. To investigate this, we performed immunoprecipitation followed by mass spectrometry (IP-MS) to identify NSP13-interacting proteins in the presence or absence of YAP in HEK293T cell nuclei (*Figure 4F* and *Source data 1*). Our analyses identified hundreds of candidate proteins that interact with NSP13 in the nucleus (*Figure 4G* and *Source data 2*), which were further analyzed by STRING-based network analysis (*Figure 4—figure supplement 6* and *Source data 3*). Gene Ontology (GO) analysis indicated that the largest clusters are involved in RNA polymerase II transcription termination, chromatin remodeling, and protein folding (*Figure 4H*). To validate these candidate interactors, we evaluated several proteins—CCT3, SMARCD1, EIF4A1, LMNA, TTF2, and YY2—and conducted co-IP assays. Whereas TEAD4 strongly interacts with NSP13 (*Figure 4—figure supplement 7*), the candidate repressors exhibited relatively weak binding to NSP13, suggesting that NSP13 associates with these proteins indirectly, potentially through a larger multiprotein complex or via chromatin-dependent interactions (*Figure 4J*).

Next, we evaluated the functional consequence of siRNA knockdown of selected candidates in YAP transactivation (*Figure 4—figure supplement 8*). Reporter assays revealed that knockdown of CCT3 and TTF2 increased endogenous YAP transactivation in HeLa cells (*Figure 4I*, *Figure 4—figure supplement 9*). To confirm the function of CCT3, we performed quantitative PCR analysis of classical

**Figure 3.** NSP13 helicase activity is required for suppressing YAP activity. (**A**) Conserved amino acid sequences of NSP13 among coronaviruses. (**B**) SARS-CoV-2 NSP13 mutant plasmids were constructed to examine YAP suppression mechanisms. NSP13-R567A, which loses its ATP consumption ability, did not inhibit YAP5SA transactivation, whereas NSP13 K345A/K347A, which loses its nucleic acid binding activity, mildly promoted YAP5SA transactivation (n=3 independent experiments; data are reported as the mean ± SD). \*\*p<0.01, \*\*\*\*p<0.0001, one-way ANOVA. (**C**) Six NSP13 truncations were constructed based on the NSP13 domain map. (**D**) Reporter assay: none of the truncations led to a reduction in YAP transactivation, and the NSP13 DNA binding domains 1A and 2A slightly increased YAP5SA activation, suggesting that the full-length NSP13 with helicase activity may be required for suppression of YAP transactivation (n=3 independent experiments; data are reported as the mean ± SD). \*p<0.05, \*\*\*\*p<0.0001, one-way ANOVA. (**E**) Summary of NSP13 mutants from SARS-CoV-2 variants. (**F**) HOP-flash reporter assay: NSP13 mutations did not affect suppression of YAP5SA transactivation (n=3 independent experiments; data are reported as mean ± SD). \*\*\*\*p<0.0001, one-way ANOVA.

The online version of this article includes the following source data and figure supplement(s) for figure 3:

**Source data 1.** Original files for western blot analysis displayed in *Figure 3B*.

**Source data 2.** Original western blots for *Figure 3B*, indicating the relevant bands and treatments.

**Source data 3.** Original files for western blot analysis displayed in *Figure 3D*.

**Source data 4.** Original western blots for *Figure 3D*, indicating the relevant bands and treatments.

**Source data 5.** Original files for western blot analysis displayed in *Figure 3F*.

**Source data 6.** Original western blots for *Figure 3F*, indicating the relevant bands and treatments.

**Figure supplement 1.** SARS-CoV-2 NSP13 mutations: structural locations and functional effect on YAP5SA transactivation.



**Figure 4.** NSP13 inactivates the YAP/TEAD4 complex by recruiting YAP repressors. (**A**) Immunofluorescence imaging showing that NSP13 colocalized with YAP5SA in cardiomyocytes of YAP5SA transgenic mice 3 days after tamoxifen injection. Scale bar, 20 μm. (**B**) Co-IP: NSP13 interacts with TEAD4, a major binding partner of YAP, in the nucleus. (**C**) Co-IP of HEK293T nuclei: NSP13 does not disrupt the interaction between YAP and TEAD4, whereas TEAD4 promotes the interaction between YAP and NSP13. (**D, E**) Immunofluorescence imaging and western blot analysis reveal that NSP13 protein

*Figure 4 continued on next page*

*Figure 4 continued*

levels increase after YAP5SA expression is induced in YAP5SA mouse cardiomyocytes. Scale bar, 50 μm. (**F**) Workflow of IP-MS. (**G**) IP-MS in nuclei (IP: NSP13), suggesting NSP13 interacts with proteins with or without YAP co-expression. Significance Analysis of INTeractome (SAINT), AvgP >0.6 are labeled in red. (**H**) GO analysis in subclusters of NSP13 interacting proteins (SAINT, AvgP >0.6, labeled with red in Figure S4C). (**I**) HOP-flash reporter assay: endogenous YAP activity is increased after the siRNA knockdown of CCT3 and TTF2 in HeLa cells (n=3 independent experiments; data are reported as mean ± SD). ***p<0.001, ****p<0.0001, one-way ANOVA. (**J**) Working model for NSP13 regulation of YAP/TEAD.

The online version of this article includes the following source data and figure supplement(s) for figure 4:

**Source data 1.** Original files for western blot analysis displayed in *Figure 4B*.

**Source data 2.** Original western blots for *Figure 4B*, indicating the relevant bands and treatments.

**Source data 3.** Original files for western blot analysis displayed in *Figure 4C*.

**Source data 4.** Original western blots for *Figure 4C*, indicating the relevant bands and treatments.

**Source data 5.** Original files for western blot analysis displayed in *Figure 4E*.

**Source data 6.** Original western blots for *Figure 4E*, indicating the relevant bands and treatments.

**Figure supplement 1.** Western blot analysis showed that neither wild-type nor mutant NSP13 directly interacted with YAP5SA in HEK 293T cells.

**Figure supplement 1—source data 1.** Original files for western blot analysis displayed in *Figure 4—figure supplement 1*.

**Figure supplement 1—source data 2.** Original western blots for *Figure 4—figure supplement 1*, indicating the relevant bands and treatments.

**Figure supplement 2.** TEAD4 domain mapping experiments showing that both the N-terminus and C-terminus are required for the interaction with NSP13.

**Figure supplement 2—source data 1.** Original files for western blot analysis displayed in *Figure 4—figure supplement 2*.

**Figure supplement 2—source data 2.** Original western blots for *Figure 4—figure supplement 2*, indicating the relevant bands and treatments.

**Figure supplement 3.** Co-immunoprecipitation in nucleus indicated that NSP13 wild-type and R567A did not disrupt YAP and TEAD4 interaction.

**Figure supplement 3—source data 1.** Original files for western blot analysis displayed in *Figure 4—figure supplement 3*.

**Figure supplement 3—source data 2.** Original western blots for *Figure 4—figure supplement 3*, indicating the relevant bands and treatments.

**Figure supplement 4.** Immunofluorescence imaging showing that both NSP13 WT and R567A restricted YAP5SA in cell nucleus.

**Figure supplement 5.** Western blot analysis of co-IP showed that nucleic acid removal did not disrupt the NSP13–TEAD4 interaction.

**Figure supplement 5—source data 1.** Original files for western blot analysis displayed in *Figure 4—figure supplement 5*.

**Figure supplement 5—source data 2.** Original western blots for *Figure 4—figure supplement 5*, indicating the relevant bands and treatments.

**Figure supplement 6.** STRING analysis of NSP13 interacting proteins.

**Figure supplement 7.** Co-IP assays indicated that the NSP13 interactors from mass spectrometry had weaker binding with NSP13 compared to TEAD4.

**Figure supplement 7—source data 1.** Original files for western blot analysis displayed in *Figure 4—figure supplement 7*.

**Figure supplement 7—source data 2.** Original western blots for *Figure 4—figure supplement 7*, indicating the relevant bands and treatments.

**Figure supplement 8.** siRNA knockdown efficiency for genes in HeLa cells, determined by using quantitative polymerase chain reaction (n=3 independent experiments; data are presented as mean ± SD).

**Figure supplement 9.** Reporter assay (8xGTIIC) results in HeLa cells revealed that endogenous YAP activity was increased after the siRNA-mediated knockdown of CCT3 and TTF2.

**Figure supplement 10.** Quantitative PCR analysis indicated increased expression of *CTGF* and *CYR61* following *CCT3* knockdown.

YAP target genes and observed increased expression of *CTGF* and *CYR61* following *CCT3* knockdown (*Figure 4—figure supplement 10*). These findings support a model where NSP13 suppresses YAP activity by recruiting suppressors to the YAP/TEAD4 complex.

## Discussion

In this study, we observed that SARS-CoV-2 infection in human lung and hiPSC-CMs reduces YAP transcriptional activity, wherein the SARS-CoV-2 helicase NSP13 significantly inhibited YAP activity. Mechanistically, NSP13 directly interacts with TEAD4 to form a YAP/TEAD4/NSP13 complex in the nucleus, which recruits suppressors such as TTF2 and CCT3 to repress YAP-TEAD transcriptional activity (*Figure 4J*).

Our data reveal a helicase-dependent inhibition of YAP by NSP13, where the K131, K345/K347, and R567 NSP13 residues are required. Based on published structural and biochemical studies, each of these residues uniquely supports helicase function: Substituting K131 with alanine (K131A) severely

reduces helicase efficiency; K345/K347 are key DNA-binding residues as mutating both (K345A/K347A) abolishes DNA binding; mutation of the ATP hydrolysis residue R567 (R567A) disables DNA unwinding. As illustrated in *Figure 4J*, NSP13 must bind DNA and hydrolyze ATP to unwind nucleic acids. This helicase-dependent process likely enables NSP13 to remodel chromatin by binding TEAD and organizing YAP repressors at the YAP/TEAD complex to prevent YAP/TEAD transactivation. In support of this mechanism, the K345A/K347A mutant, unable to anchor to DNA, fails to repress YAP as YAP-driven transcription is slightly increased in this mutant (*Figure 3B*). Likewise, the ATPase-dead R567A can bind DNA but does not unwind and remodel chromatin to recruit YAP repressors, resulting in a loss of YAP suppression (*Figure 3B and F*). Our model demonstrates that both DNA binding and ATP-dependent unwinding are essential for NSP13 to suppress YAP transcriptional activity.

TEAD transcription factors, the major partners of YAP, are the primary nuclear effectors of Hippo-YAP signaling and play critical roles in cancer development (*Huh et al., 2019*; *Pobbati et al., 2023*). We observed that NSP13 interacts with TEAD4, but does not disrupt the YAP/TEAD4 interaction, suggesting a novel regulatory mechanism for TEAD. IP-MS revealed that NSP13-interacting proteins such as TTF2 and CCT3 suppress the YAP-TEAD4 complex. TTF2, an SWI2/SNF2 family member, facilitates the removal of RNA polymerase II from the DNA template via ATP hydrolysis (*Liu et al., 1998*; *Jiang et al., 2004*). Importantly, termination of transcription complex elongation by TTF2 appears minimally affected by template position (*Jiang et al., 2004*). CCT3 is a component of the chaperonin-containing T-complex, a molecular chaperone complex that promotes protein folding following ATP hydrolysis. Previous reports revealed that the liver cancer biomarker CCT3 (*Liu et al., 2019*) positively regulates YAP protein stability. We found that CCT3 interacts with NSP13 and inhibits YAP transactivation. Moreover, CCT3 knockdown increases YAP target gene expression, suggesting a context-dependent role of CCT3 in YAP regulation. While Co-IP assays detected weak interactions between NSP13 and CCT3 or TTF2, a strong NSP13/TEAD4 interaction was evident. These data suggest that NSP13 associates with CCT3 and TTF2 indirectly as part of a multiprotein complex in a chromatin structure-dependent manner, although further study is required to elucidate these mechanisms.

In conclusion, our study reveals a novel function of NSP13 in suppressing the YAP/TEAD transcriptional complex. These findings advance our understanding of the pathological effects of SARS-CoV-2 on host cells. Given that NSP13 directly interacts with TEAD4 and strongly suppresses YAP transactivation, this work provides a new avenue for modulating YAP activity as a potential treatment for YAP-driven diseases.

# Materials and methods
## Mice
Mouse studies were performed in accordance with the National Institutes of Health Guide for the Care and Use of Laboratory Animals and with approval from the Institutional Animal Care and Use Committee at Baylor College of Medicine. We used αMyHC-MerCreMer mice (The Jackson Laboratory, strain #005657, RRID:IMSR_JAX:005657), wild-type (WT) mice (The Jackson Laboratory, strain #000664, RRID:IMSR_JAX: 000664), and αMyHC-MerCreMer; YAP5SA mice (a mixed genetic background of C57BL/6 and 129SV) in this study. The YAP5SA transgenic mice were constructed by Martin lab (*Monroe et al., 2019*). Animals were assigned to experimental groups using simple randomization. The AAV9 virus (a total of $5 \times 10^{11}$ viral genomes, 120 µl total volume) was delivered by retro-orbital injection 2 weeks before tamoxifen injection. For the survival experiment in *Figure 2*, two low doses of tamoxifen (10 µg/g) were administered to 6-week-old mice by intraperitoneal injection. The mouse cardiac function was evaluated by echocardiography at days 4 and 8 post-tamoxifen injection, and all the surviving mice were sacrificed at day 21. For the experiments in *Figure 4*, two tamoxifen doses were injected (*Figure 4A* 10 µg/g; *Figure 4D and E* 50 µg/g), and mouse hearts were collected at day 3 post-tamoxifen injection. The influence of sex was analyzed as a source of variation. For all mouse studies, we examined equal numbers of male and female animals, and similar findings are reported for both sexes. All analyses in mice were performed blinded.

## RNA-seq analysis
To analyze lung samples from COVID-19 patients, snRNA sequencing data of 7 control lungs and 35 COVID-19 lungs from GSE171668 (*Delorey et al., 2021*) and GSE171524 (*Melms et al., 2021*) were

downloaded and analyzed using the Seurat v4 software suite (*Hao et al., 2021*). A total of 223,106 nuclei were used in the analysis. Each sample was normalized and batch corrected using SCTrans-formation. Mitochondrial percentage was used to regress out technical variability between batches, and we used harmony (*Korsunsky et al., 2019*) integration to remove technical variation among samples. The YAP score was evaluated using 38 canonical YAP target genes. For analyzing iPSC-CMs infected with SARS-CoV-2, bulk RNA-seq data were obtained from *Perez-Bermejo et al., 2021*. With this dataset, we reanalyzed and evaluated the average expression of 302 cardiomyocyte-specific YAP target genes (*Monroe et al., 2019*).

## Expression plasmids

Expression plasmids encoding HA-tagged WT, mutant, or truncated NSP13 were generated by polymerase chain reaction (PCR) and subcloned into pXF4H (N-terminal HA tag) derived from pRK5 (Genetech). Myc-tagged WT or truncated TEAD4, flag-tagged WT YAP, flag-tagged YAP5SA, and Myc-tagged NSP13 interacting proteins, including CCT3, SMARCD1, EIF4A1, LMNA, TTF2, and YY2, were generated by PCR and subcloned into a pcDNA3 backbone. HA-tagged WT LATS2, LATS2 KR, and pRL-TK_Luc were gifts from Dr. Pinglong Xu's lab. Reporters of HOP-flash (#83467) and 8xGTIIC-luciferase (#34615) were purchased from Addgene. The NSP plasmids used in YAP transactivation screening experiments were gifts from Dr. Nevan J. Krogan's lab. All plasmids were confirmed by performing DNA sequencing.

## Cell culture and transfection

HEK293T and HeLa cells were cultured in Dulbecco's Modified Eagle Medium with 10% fetal bovine serum. Lipofectamine 2000 (Thermo Fisher) or Lipofectamine 3000 (Thermo Fisher) reagents were used for plasmid transfection. LipofectAmine RNAiMAX (Thermo Fisher) was used for siRNA trans-fection. Both cell lines were obtained from the American Type Culture Collection (ATCC, HEK293T CRL-3216, HeLa CCL-2). The identity of the cell lines was authenticated by short tandem repeat (STR) profiling, and all cell lines tested negative for mycoplasma contamination using PCR-based assays. HeLa cells are included on the list of commonly misidentified cell lines maintained by the International Cell Line Authentication Committee (ICLAC). However, their use in this study was intentional and appropriate for the experimental design. The specific characteristics of HeLa cells were required for our assays, and their identity and relevance to the research objectives were confirmed through STR profiling and functional validation.

## Immunoprecipitation-mass spectrometry, database search, and data analysis

HEK293T cells were plated on a 150 mm tissue culture dish using 10% FBS DMEM 24 h post-plating. Cells were transfected with Flag-YAP1, HA-NSP13, or Flag-YAP1 + HA-NSP13 by Lipofectamine 3000 using standard manufacturer's protocols. Each experimental group was performed in duplicate to ensure reproducibility and reliability. 24 h post-transfection, cells were collected in a hypotonic lysis buffer (5 mM NaCl, 20 mM HEPES, pH 7.5, 0.4% NP-40) supplemented with protease inhibitor (Roche) and phosphatase inhibitor (Roche). Cell suspensions were drawn into a 3 ml syringe using a 25 G needle and expelled in one rapid stroke into a new tube. After three rounds of lysis, the suspension was centrifuged at $600 \times g$ 4°C for 10 min to pellet nuclei. The supernatant was collected in a different tube, and the nuclear fraction was washed in the buffer twice. After the final wash, nuclei were resus-pended in 20 mM HEPES, 150 mM NaCl, 2.5 mM $MgCl_2$, 0.1% NP 40, and nuclear fractions were soni-cated using Bioruptor Pico for 10 cycles of 30 s on and 30 s off. Subsequently, lysate was centrifuged for 20 min at $2 \times 1000$ $g$ at 4°C. The supernatant was incubated with HA magnetic beads (Thermo Fisher) for 2 h at 4°C with continuous rotation. Antibody and protein-bound beads were washed thrice for 15 min in lysis buffer at 4°C with constant rotation. Washed beads were boiled in 40 µl of 1X NUPAGE LDS sample buffer (Invitrogen) and subjected to SDS-PAGE (NuPAGE 10% Bis-Tris Gel, Invit-rogen). Eluted proteins were visualized with Coomassie Brilliant blue stain and excised into gel pieces according to molecular size. Individual gel pieces were destained and subjected to in-gel digestion using trypsin (GenDepot T9600). Tryptic peptides were resuspended in 10 µl of loading solution (5% methanol containing 0.1% formic acid) and subjected to nanoflow LC-MS/MS analysis with a nano-LC 1000 system (Thermo Scientific) coupled to Orbitrap Elite (Thermo Scientific) mass spectrometer.

Peptides were loaded onto a Reprosil-Pur Basic C18 (1.9 μm, Dr. Maisch GmbH, Germany) precolumn of 2 cm × 100 μm size. The precolumn was switched in-line with an in-house 50 mm × 150 μm analytical column packed with Reprosil-Pur Basic C18 equilibrated in 0.1% formic acid/water. Peptides were eluted using a 75 min discontinuous gradient of 4–26% acetonitrile/0.1% formic acid at a 700 nl/min flow rate. Eluted peptides were directly electro-sprayed into Orbitrap Elite mass spectrometer operated in the data-dependent acquisition mode acquiring fragmentation spectra of the top 25 strongest ions and under the direct control of Xcalibur software (Thermo Scientific).

MS/MS spectra were analyzed using the target-decoy mouse RefSeq database in the Proteome Discoverer 1.4 interface (Thermo Fisher), employing the Mascot algorithm (Mascot 2.4, Matrix Science). The precursor mass tolerance was set to 20 ppm, with a fragment mass tolerance of 0.5 daltons and a maximum of two allowed missed cleavages. Dynamic modifications included oxidation, protein N-terminal acetylation, and destreaking. Peptides identified from the Mascot results were validated at a 5% false discovery rate, and the iBAQ algorithm was applied to assess protein abundance, enabling comparisons of relative quantities among different proteins in the sample. The iBAQ value was determined by normalizing the summed peptide intensity by the number of theoretically observable tryptic peptides for each protein. PSM values from Proteome Discover were uploaded to the SAINT user interface on the CRAPOME website for SAINT analysis. SAINT probability and fold change were calculated using YAP IP as a user-defined control.

## Luciferase reporter assay

HEK293T or HeLa cells were transfected with WT YAP- or YAP5SA-responsive HOP-flash plasmid or 8xGTIIC-luciferase reporter, which has an open-reading frame encoding firefly luciferase, along with the pRL-Luc with *Renilla* luciferase coding as the internal control for transfection and other expression vectors (NSPs), as specified. After 24 h of transfection with the indicated treatments, cells were lysed with passive lysis buffer (Promega). Luciferase assays were performed by using a dual-luciferase assay kit (Promega), and data were quantified with POLARstar Omega (BMG Labtech) and normalized to the internal *Renilla* luciferase control.

## AAV9 viral packaging

Viral vectors were used as previously described (*Leach et al., 2017*). The construct containing HA-tagged NSP13 was cloned into the pENN.AAV.cTNT, p1967-Q vector (AAV9-HA-NSP13). Empty vector-encoding green fluorescent protein was used as the control (AAV9-GFP). Both vectors were packaged into the muscle-trophic serotype AAV9 by the Intellectual and Developmental Disabilities Research Center Neuroconnectivity Core at Baylor College of Medicine. After being titered, viruses were aliquoted (1 × 10$^{13}$ viral genome particles per tube), immediately frozen, and stored long-term at −80°C. Each aliquot was diluted in saline to make a 120 ul injection solution.

## Echocardiography

For cardiac function analysis, echocardiography was performed on a VisualSonics Vevo 2100 system with a 550 s probe. B-mode images and M-mode images were captured on a short-axis projection. Ejection fraction, fractional shortening, diameter of diastolic left ventricle, diameter of systolic left ventricle, and end-diastolic volume were calculated by using a cardiac measurement package installed in the Vevo2100 system.

## Histology and immunofluorescence staining

Hearts were fixed in 4% paraformaldehyde overnight at 4°C, dehydrated in serial ethanol and xylene solutions, and embedded in paraffin. For immunofluorescence staining, slides were sectioned at 7 μm intervals. For paraffin sections, samples were deparaffinized and rehydrated, treated with 3% H$_2$O$_2$ in EtOH and then with antigen retrieval solution (Vector Laboratories Inc, Burlingame, CA, USA), blocked with 10% donkey serum in phosphate-buffered saline, and incubated with primary antibodies. Antibodies used were rabbit anti-HA (#3724, Cell Signaling, RRID:AB_1549585) and rat anti-flag (NBP1-06712, Novus Biologicals, RRID:AB_1625981). Immunofluorescence-stained images were captured on a Zeiss LSM 780 NLO Confocal/2-photon microscope.

## Statistical analysis

Sample sizes for mice experiments were determined based on power calculations using preliminary data, with the goal of achieving at least 80% statistical power at a significance level of 0.05. Mice were

randomly assigned to experimental groups. For *in vitro* studies, samples were randomly distributed across treatment groups based on initial cell density or viability. Investigators were blinded to group assignments during data collection and analysis to reduce potential bias. Inclusion and exclusion criteria were pre-established before the experiments. Animals or samples were excluded from analysis if they exhibited procedural complications (e.g., unexpected mortality unrelated to the intervention, or technical failure such as injection leakage). The number of biological replicates (n), statistical tests used, and significance thresholds are indicated in the figure legends and throughout the manuscript.

For the data in this article, the specific statistical test used is presented in the figure legend. In snRNA-seq analyses, Yap score differences between AT1 from Control and COVID-19 patients were identified via the Wilcoxon test. For analyzing iPSC-CMs infected with SARS-CoV-2, the expression score of cardiomyocyte YAP targets was evaluated by Student's *t*-test. Mouse survival rates in *Figure 2D* were analyzed by the log-rank (Mantel–Cox) test. Statistical significances were evaluated using three-way ANOVA and Šídák's multiple comparisons test for other cardiac data, including EF, FS, and systolic LV diameter. For reporter assays in cells and heart/body ratio in *Figure 2—figure supplement 6*, the statistical significance of the observed differences in mean was evaluated using a one-way or two-way ANOVA and the post hoc Tukey's multiple comparisons test. p-Values <0.05 were considered statistically significant (*p<0.05, **p<0.01, ***p<0.001, ****p<0.0001).

## Acknowledgements

We are grateful to Drs. Nevan J Krogan and Pinglong Xu for the plasmid gifts. Dr. Hongxin Guan assisted with NSP13 mutant structure analysis. Rebecca Bartow, PhD, of the Department of Scientific Publications at The Texas Heart Institute, provided editorial support. National Institutes of Health grant HL 127717 (JFM). National Institutes of Health grant HL 130804 (JFM). National Institutes of Health grant HL 118761 (JFM). Vivian L Smith Foundation (JFM). 2020 COVID-19 RFA (JFM). AHA postdoctoral fellowship 903411 (FM). AHA Career Development Award 24CDA1274610 (FM). AHA postdoctoral fellowship 903651 (RL). K99 HL169742 (JS). Don McGill Gene Editing Laboratory of The Texas Heart Institute (XL).

## Additional information

### Competing interests

James F Martin: founder and owns shares in Yap Therapeutics, and is a co-inventor on the following patents associated with this study: patent no. US20200206327A1 entitled "Hippo pathway deficiency reverses systolic heart failure post-infarction," patent no.15/642200.PCT/US2014/ 069349 101191411 entitled "Hippo and dystrophin complex signaling in cardiomyocyte renewal," and patent no. 15/102593.PCT/US2014/069349 9732345 entitled "Hippo and dystrophin complex signaling in cardiomyocyte renewal.". The other authors declare that no competing interests exist.

### Funding

| Funder | Grant reference number | Author |
|---|---|---|
| National Heart, Lung, and Blood Institute | HL127717 | James F Martin |
| National Heart, Lung, and Blood Institute | HL130804 | James F Martin |
| National Heart, Lung, and Blood Institute | HL118761 | James F Martin |
| American Heart Association | 903411 | Fansen Meng |
| American Heart Association | 903651 | Rich G Li |
| National Heart, Lung, and Blood Institute | K99 HL169742 | Jeffrey D Steimle |

| Funder | Grant reference number | Author |
| --- | --- | --- |
| American Heart Association | 10.58275/AHA. 24CDA1274610.pc.gr. 193655 | Fansen Meng |

The funders had no role in study design, data collection and interpretation, or the decision to submit the work for publication.

## Author contributions

Fansen Meng, Conceptualization, Data curation, Formal analysis, Supervision, Funding acquisition, Validation, Investigation, Methodology, Writing – original draft, Project administration, Writing – review and editing; Jong Hwan Kim, Data curation, Formal analysis; Chang-Ru Tsai, Formal analysis, Validation; Jeffrey D Steimle, Formal analysis, Methodology; Jun Wang, Yufeng Shi, Vaibhav Deshmukh, Validation, Methodology; Rich G Li, Investigation; Bing Xie, Shijie Liu, Validation; Xiao Li, Investigation, Methodology; James F Martin, Conceptualization, Resources, Project administration, Writing – review and editing

## Author ORCIDs

Fansen Meng ⓘ https://orcid.org/0000-0003-3173-5512
Xiao Li ⓘ https://orcid.org/0000-0001-8527-4438
James F Martin ⓘ https://orcid.org/0000-0002-7842-9857

Reviewer #1 (Public review): https://doi.org/10.7554/eLife.100248.3.sa1
Reviewer #2 (Public review): https://doi.org/10.7554/eLife.100248.3.sa2
Author response https://doi.org/10.7554/eLife.100248.3.sa3

# Additional files

## Supplementary files

MDAR checklist

Source data 1. IP-MS raw data.

Source data 2. IP-MS analysis.

Source data 3. STRING analysis.

Source data 4. Supporting data values in figures.

## Data availability

All the sequencing data in Figure 1 and Figure 1—figure supplements 1–4 were downloaded from two published papers (GSE171668 and GSE171524). Source data 4 (Supporting data values excel) contains the numerical data used to generate the figures. Source datas 1–3 provide raw data and analysis of mass spectrum.

The following previously published datasets were used:

| Author(s) | Year | Dataset title | Dataset URL | Database and Identifier |
|---|---|---|---|---|
| Delorey TM, Ziegler CGK, Heimberg G, Normand R, Yang Y, Segerstolpe A, Abbondanza D, Fleming SJ, Subramanian A, Montoro DT, Jagadeesh KA, Dey KK, Sen P, Slyper M, Pita-Juarez YH, Phillips D, Biermann J, Bloom-Ackermann Z, Barkas N, Ganna A, Gomez J, Melms JC, Katsyv I, Normandin E, Naderi P, Popov YV, Raju SS, Niezen S, Tsai LT, Siddle KJ, Sud M, Tran VM, Vellarikkal SK, Wang Y, Amir-Zilberstein L, Atri DS, Beechem J, Brook OR, Chen J, Divakar P, Dorceus P, Engreitz JM, Essene A, Fitzgerald DM, Fropf R, Gazal S, Gould J, Grzyb J, Harvey T, Hecht J, Hether T, Jane-Valbuena J, Leney-Greene M, Ma H, McCabe C, McLoughlin DE, Miller EM, Muus C, Niemi M, Padera R, Pan L, Pant D, Pe'er C, Pfiffner-Borges J, Pinto CJ, Plaisted J, Reeves J, Ross M, Rudy M, Rueckert EH, Siciliano M, Sturm A, Todres E, Waghray A, Warren S, Zhang S, Zollinger DR, Cosimi L, Gupta RM, Hacohen N, Hibshoosh H, Hide W, Price AL, Rajagopal J, Tata PR, Riedel S, Szabo G, Tickle TL, Ellinor PT, Hung D, Sabeti PC, Novak R, Rogers R, Ingber DE, Jiang ZG, Juric D, Babadi M, Farhi SL, Izar B, Stone JR, Vlachos IS, Solomon IH, Ashenberg O, Porter CBM, Li B, Shalek AK, Villani AC, Rozenblatt-Rosen O, Regev A | 2021 | COVID-19 tissue atlases reveal SARS-COV-2 pathology and cellular targets | https://www.ncbi.nlm.nih.gov/geo/query/acc.cgi?acc=GSE171668 | NCBI Gene Expression Omnibus, GSE171668 |

*Continued on next page*

*Continued*

| Author(s) | Year | Dataset title | Dataset URL | Database and Identifier |
|---|---|---|---|---|
| Melms JC, Biermann J, Huang H, Wang Y, Nair A, Tagore S, Katsyv I, Rendeiro AF, Amin AD, Schapiro D, Frangieh CJ, Luoma AM, Filliol A, Fang Y, Ravichandran Y, Clausi MG, Alba GA, Rogava M, Chen SW, Ho P, Montoro DT, Kornberg AE, Han AS, Bakhoum MF, Anandasabapathy N, Suarez-Farinas M, Bakhoum SF, Bram Y, Borczuk A, Guo XV, Lefkowitch JH, Marboe C, Lagana SM, Del Portillo A, Tsai EJ, Zorn E, Markowitz GS, Schwabe RF, Schwartz RE, Elemento O, Saqi A, Hibshoosh H, Que J, Izar B | 2021 | Columbia University/NYP COVID-19 Lung Atlas | https://www.ncbi.nlm.nih.gov/geo/query/acc.cgi?acc=GSE171524 | NCBI Gene Expression Omnibus, GSE171524 |

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
