## [Editor Report · eLife Assessment]

This **important** study elucidates the molecular function of the SARS-CoV-2 helicase NSP13, which inhibits the transcriptional activity of the YAP/TEAD complex *in vitro* and *in vivo*. The evidence supporting the authors' claims is **compelling**, based on cell biological assays and multi-omics studies. This work contributes to the understanding of the new regulatory mechanism of YAP/TEAD after SARS-CoV-2 infection and will be of interest to researchers investigating COVID-19 infection and the Hippo-YAP signaling pathway.

---

## [Referee Report · Reviewer #1 (Public review)]

In the revised manuscript, Meng et al. report that SARS-CoV-2 infection suppresses YAP target gene transcription in both patient lung samples and iPSC-derived cardiomyocytes. Among the tested viral proteins, the helicase nonstructural protein 13 (NSP13) was identified as a key factor that impairs YAP/TEAD transcriptional activity. Through mutagenesis and protein-protein interaction studies, the authors propose a mechanism where NSP13 binds YAP/TEAD complex, remodels chromatin structure, and recruits transcriptional repressors to inhibit YAP/TEAD's transcriptional activity.

Overall, this study uncovers a novel regulation of Hippo signaling by SARS-CoV-2 through NSP13, suggesting a potential role of this growth-related pathway in host innate immune response to viral infection. While these findings are intriguing, future studies are needed to validate the involvement of YAP/TEAD in patient tissues and to assess their potential as therapeutic targets against SARS-CoV-2.

---

## [Referee Report · Reviewer #2 (Public review)]

Summary:

The manuscript by Meng et al. describes a role for the coronavirus helicase NSP13 in the regulation of YAP-TEAD-mediated transcription. The authors present data that NSP13 expression in cells reduces YAP-induced TEAD luciferase reporter activity and that NSP13 transduction in cardiomyocytes blocks hyperactive YAP-mutant phenotypes *in vivo*. Mechanisms by which viral proteins (particularly those from coronaviruses) intersect with cellular signaling events is an important research topic, and the intersection of NSP13 with YAP-TEAD transcriptional activity (independent of upstream Hippo pathway mediated signals) offers new knowledge that is of interest to a broad range of researchers.

Strengths:

The manuscript presents convincing data mapping the effects of NSP13 on YAP-TEAD reporter activity to the helicase domain. Moreover, the *in vivo* data demonstrating that NSP13 expression in YAP5SA mouse cardiomyocytes increased survival animal rates, and restored cardiac function is striking and is supportive of the model presented.

Weaknesses:

While there are some hints at the mechanisms by which NSP13 regulates YAP-TEAD activity through the identification of NSP13-associated proteins by mass spec, the relationships and functions of these factors in the context of YAP-TEAD regulation requires further study in the future.

---

## [Author Response]

The following is the authors’ response to the original reviews.

**Reviewer #1 (Public Review):**
Major points(1) The authors discovered a novel regulation of the Hippo-YAP pathway by SARS-CoV-2 infection but did not address the pathological significance of this finding. It remains unclear why YAP downstream gene transcription needs to be inhibited in response to SARS-CoV-2 infection. Is this inhibition crucial for the innate immune response to SARS-CoV-2? The authors should re-analyze their snRNA-seq and bulk RNA-seq data described in Figure 1 to determine whether any of the affected YAP downstream genes are involved in this process.

We appreciate the reviewer’s suggestion to clarify the pathological significance of YAP pathway inhibition in SARS-CoV-2 infection. To address this, we re-analyzed our snRNA-seq and bulk RNA-seq datasets to determine whether YAP target genes overlap with known mediators of the innate immune response. As described in Fig. 1C, bulk RNA-seq revealed decreased expression of multiple YAP downstream targets linked to innate immune regulation (e.g., Thbs1, Ccl2, Axl, and Csf1) in SARS-CoV-2–infected cells *in vitro*.

snRNA-seq of alveolar type I (AT1) cells from COVID-19 patients revealed a more complex landscape: While we observed reduced YAP activity overall (Fig. 1G), multiple YAP target genes involved in innate immunity and cytokine signaling were paradoxically elevated (Supplemental Fig. 1E). Several factors likelt explain these conflicting observations: 1. In the lung, AT1 cells (which are critical for gas exchange) may cell specifically respond to virus infection by upregulating genes related to immune response by other signaling pathway(s); 2. *In vivo*, SARS-CoV-2 infection triggers a surge in cytokines, chemokines, and other local factors that can differentially modulate YAP binding sites and thus affect its downstream targets, a complexity not fully captured *in vitro*; 3. YAP is highly sensitive to mechanical signals and tissue architecture. The 3D structure of altered cell–cell junctions in infected lung tissue, and fluid shear stress in the alveolar space could shape YAP target gene transcription differently from simplified monolayer cell cultures.

We have expanded the results section of the new version to include the above points. We also acknowledge that ongoing and future work is needed to delineate the exact molecular and tissue-specific pathways through which YAP inhibition confers a potential advantage in combating SARS-CoV-2.

(2) The authors concluded that helicase activity is required for NSP13-induced inhibition of YAP transcriptional activity based on mutation studies (Figure 3B). This finding is somewhat confusing, as K131, K345/K347, and R567 are all essential residues for NSP13 helicase activity while mutating K131 did not affect NSP13's ability to inhibit YAP (Figure 3B). Additionally, there are no data showing exactly how NSP13 inhibits the YAP/TEAD complex through its helicase function. This point was also not reflected in their proposed working model (Figure 4H).

We appreciate the reviewer’s concerns regarding the helicase‐dependent inhibition of YAP by NSP13, particularly the roles of K131, K345/K347, and R567. Based on published structural and biochemical studies, each of these residues uniquely supports helicase function (1): K131 is crucial for stabilizing the NSP13 stalk region by interacting with S424. Substituting K131 with alanine (K131A) reduces helicase efficiency but does not completely abolish it; K345/K347 are key DNA‐binding residues, and mutating both (K345A/K347A) largely prevents NSP13 from binding DNA, thus eliminating unwinding. R567 is critical for ATP hydrolysis, and the R567A mutant retains DNA binding capacity but fails to unwind it. In Fig. 3B, K131A suppresses YAP transactivation to nearly the same extent as wild‐type NSP13, suggesting that partial helicase activity is sufficient for complete YAP/TEAD inhibition. Conversely, the K345A/K347A and R567A mutants show markedly diminished repression, underscoring the importance of DNA binding and ATP hydrolysis.

As the new Fig. 4J illustrates, NSP13 must bind DNA and hydrolyze ATP to unwind nucleic acids. This helicase‐dependent process likely enables NSP13 to remodel chromatin structure by binding TEAD and properly organize YAP repressors at YAP/TEAD complex to prevent YAP/TEAD transactivation. In support of this mechanism, the K345A/K347A mutant, unable to anchor to DNA, fails to repress YAP and slightly increases YAP‐driven transcription (Fig. 3B), presumably by mislocalizing YAP repressors. Likewise, the ATPase‐dead R567A can bind DNA but does not unwind and remodel chromatin to recruit YAP repressors, resulting in a loss of YAP suppression (Fig. 3B and 3F). Our revised model demonstrates that both DNA binding and ATP‐dependent unwinding are essential for NSP13 to suppress YAP transcriptional activity. We have updated the results, discussion, and model accordingly.

(3) The proposed model that NSP13 binds TEAD4 to recruit repressor proteins and inhibits YAP/TEAD downstream gene transcription (Figure 4H) needs further characterization. Second, NSP13 is a DNA-binding protein, and its nucleic acid-binding mutant K345A/K347A failed to inhibit YAP transcriptional activity (Figure 3B). The authors should investigate whether NSP13 could bind to the TEAD binding sequence or the nearby sequence on the genome to modulate TEAD's DNA binding ability. Third, regarding the identified nuclear repressors, the authors should validate the interaction of NSP13 with the ones whose loss activates YAP transcriptional activity (Figure 4G). Lastly, why can't NSP13 bind TEAD4 in the cytoplasmic fractionation if both NSP13 and TEAD4 are detected there (Figure 3B)? This finding indicates their interaction is not a direct protein-protein interaction but is mediated by something in the nucleus, such as genomic DNA.

(1) Low TEAD expression in HEK293T cells: Our IP-MS experiments were performed in HEK293T cells, which, according to the Human Protein Atlas, express TEAD1–4 at comparatively low levels (TEAD1: 16.5, TEAD2: 16.4, TEAD3: 4.9, TEAD4: 38.7 nTPM). In contrast, HeLa cells, where we successfully validated NSP13-mediated YAP suppression (Fig. 4H, Supplementary Fig.5B-D), show higher expression of these TEAD isoforms (TEAD1: 97.1, TEAD2: 27.3, TEAD3: 12.2, TEAD4: 48.1 nTPM). Therefore, insufficient TEAD abundance in HEK293T cells may limit the sensitivity needed to detect TEAD–NSP13 interactions in our proteomic screens.

(2) Transience and potential DNA dependence: Our co-immunoprecipitation (co-IP) experiments (Fig. 4B, Supplementary Fig.4C-E) indicated that NSP13–TEAD4 binding is low-affinity. Under standard IP-MS conditions (which typically do not include chemical cross-linkers or nucleic acids to stabilize transient complexes), weak or short-lived interactions can be lost during washes or sample processing.

(3). Additional supporting evidence: We carefully checked our IP-MS data and found that the well-known TEAD binding proteins, including CTBP1/2 and GATA4, were pulled down, suggesting TEAD’s absence does not rule out an NSP13–TEAD association.

(3a) We acknowledge that our NSP13 immunoprecipitation–mass spectrometry (IP-MS) did not identify any TEAD proteins (Fig. 4G and IP-MS tables). Several factors likely contributed to this outcome:

(3b) We sincerely appreciate the reviewer’s insightful suggestion. While we agree that mapping NSP13 occupancy at individual TEAD-binding motifs is valuable, we respectfully consider this to be beyond the scope of the current study. Biochemical and structural work on coronavirus NSP13 shows that it recognizes nucleic‑acid substrates primarily through their 5′ single‑stranded overhang and duplex architecture, not through a defined base sequence(2, 3). Accordingly, our data (Fig. 3B and 3F) indicate that DNA binding ability, rather than recognition of a specific motif, enables NSP13 to perform its helicase activity in proximity to TEAD and recruit repressors. Moreover, the DNA‑binding mutant K345A/K347A and the ATPase‑dead mutant R567A both fail to suppress YAP/TEAD transcription despite retaining the ability to interact with TEAD (Fig. 3B). These loss‑of‑function phenotypes demonstrate that NSP13’s chromatin engagement and unwinding activity, rather than sequence‑restricted targeting, are essential for repression. For these reasons, motif‑specific binding assays were not pursued in this revision, but we clarified in the discussion that NSP13’s DNA engagement is likely structural or TEAD-dependent, rather than sequence‑directed. We also highlighted this as an important avenue for future investigation.

(3c) To validate the NSP13 interacting proteins from our IP-MS data, we generated plasmids expressing several candidates (CCT3, SMARCD1, EIF4A1, LMNA, TTF2, and YY2) and performed co-IP assays. As predicted, we confirmed the robust interaction between NSP13 and TEAD (Supplemental Fig. 5E). However, these putative nuclear repressors exhibited weak binding to NSP13 compared with TEAD4, suggesting that NSP13 associates with them indirectly, possibly as part of a larger multiprotein complex or depending on the chromatin structure, rather than via direct protein–protein interaction (Fig. 4J).

(3d) We appreciate the reviewer’s question. To investigate whether their association might be DNA‐dependent, we performed co‐IP experiments using nuclear lysates in the presence or absence of various nucleases: Universal Nuclease (which degrades all forms of DNA and RNA), DNase I (which cleaves both single‐ and double‐stranded DNA), and RNase H (which selectively cleaves the RNA strand in RNA/DNA hybrids). Our findings revealed that nucleic acid removal did not disrupt the NSP13/TEAD4 interaction (Supplemental Fig.4E), indicating that their binding is not solely mediated by DNA or RNA.

**Reviewer #2 (Public Review):**
Specific comments and suggestions for improvement of the manuscript:(1) NSP13 has been reported to block, in a helicase-dependent manner, episomal DNA transcription (PMID: 37347173), raising questions about the effects observed on the data shown from the HOP-Flash and 8xGTIIC assays. It would be valuable to demonstrate the specificity of the proposed effect of NSP13 on TEAD activation by YAP (versus broad effects on reporter assays) and also to show that NSP13 reduces the function of endogenous YAP-TEAD transcriptional activity (i.e., does ectopic NSP13 expression reduce the expression of YAP induced TEAD target genes in cells).

We appreciate the reviewer’s comments and have carefully revisited the conclusions from the published paper(4) (PMID: 37347173), which reported that NSP13 suppresses episomal DNA transcription, as evidenced by reduced Renilla luciferase (driven by the herpes simplex virus thymidine kinase promoter) and GFP expression upon co‐expression with NSP13. For our experiments, we used a dual‐luciferase assay with Renilla luciferase (under the same promoter) as an internal control. After re-examining our raw Renilla luciferase data (now provided in the supplemental Excel file “Supporting data value”), we found that while 100 ng of NSP13 did not affect Renilla luciferase levels, 400 ng of NSP13 reduced them by approximately 50% relative to the YAP5SA‐only group (Supplemental Fig.2B, Fig.3C-D). We observed a similar reduction with NSP13 truncation mutants—an outcome not fully consistent with the published study (Supplemental Fig.3D, PMID: 37347173). However, unlike their finding of robust episomal DNA suppression, our data indicate that the K345A/K347A mutant of NSP13, which lacks DNA‐binding ability, completely lost its suppressive effect (Fig.3B).

We performed additional Notch reporter assays to address the concern that NSP13 might nonspecifically inhibit episomal DNA transcription (including the HOP‑Flash and 8×GTIIC reporters). These experiments revealed that co‑expression of NSP13 with NICD (Notch intracellular domain) does not suppress Notch signaling (Supplemental Fig. 2C), indicating that NSP13 does not globally block all reporter systems. To evaluate whether NSP13 reduces endogenous YAP‑TEAD activity, we transiently overexpressed NSP13 WT and its R567A mutant in HeLa cells. However, bulk RNA‑seq and qPCR analyses did not reveal a clear decrease in YAP target genes, possibly due to the low transfection efficiency (< 50%, Supplemental Fig.4D). Interestingly, we observed that YAP5SA was predominantly retained in the nucleus upon NSP13 or R567A co‑expression, suggesting that NSP13 (or together with its interacting partners) restricts YAP5SA cytoplasmic shuttling. Future studies will involve stable cell lines expressing NSP13 WT or R567A to better characterize the mechanisms driving YAP5SA nuclear retention and clarify how NSP13 specifically suppresses YAP activity.

(2) While the IP-MS experiment may have revealed new regulators of TEAD activity, the data presented are preliminary and inconclusive. No interactions are validated and beyond slight changes in TEAD reporter activity following knockdown, no direct links to YAP-TEAD are demonstrated, and no link to NPS13 was shown. Also, no details are provided about the methods used for the IP-MS experiment, raising some concerns about potential false positive associations within the data.

We appreciate the reviewer’s feedback regarding our IP-MS findings and acknowledge that additional validation is required to establish definitive links between the identified putative regulators, YAP-TEAD, and NSP13. We have taken the following steps (and plan further experiments) to address these concerns:

(2a) Co-IP validation: Same with the answer for Reviewer #1 (3c), we generated plasmids expressing several top candidate interactors from the IP-MS data (CCT3, SMARCD1, EIF4A1, LMNA, TTF2, and YY2) and performed direct co-IP assays in a more controlled setting. The results indicated that these putative NSP13 interactors had weaker binding compared to TEAD4, implying that NSP13 may associate with them as part of a larger complex or depending on the chromatin structure rather than through a direct protein–protein interaction (Fig. 4J).

(2b) qPCR validation: Beyond reporter assays for evaluating YAP transactivation after the candidate YAP suppressor knockdown (Fig. 4H and Supplemental Fig. 5C), we performed qPCR to detect YAP activation on endogenous YAP-TEAD target genes (e.g., CTGF CYR61, and AMOTL2) after CCT3 knockdown. Expression of CTGF and CYR61 was higher compared to control (Supplemental Fig. 5D), strengthening the case for an interaction relevant to YAP-TEAD signaling.

(2c) To investigate how NSP13‐interacting proteins link to the YAP/TEAD complex, we examined the IP‑MS dataset and identified several well‐known YAP and TEAD binding partners, including CTBP1/2 (TEAD‐binding), GATA4 (TEAD‐binding), and multiple 14‐3‐3 isoforms (YWHAZ/YWHAB/YWHAH/YWHAQ, YAP binding). These findings suggest that NSP13 may form a larger nuclear complex with YAP/TEAD and associated cofactors. In the future, we will determine whether these putative TEAD regulators also interact with NSP13 under various conditions (e.g., in the presence or absence of DNA) and whether co‐expression of NSP13 influences their association with YAP or TEAD. This approach will clarify how NSP13 might leverage these factors to regulate YAP‐TEAD function.

(2e) For the mass spectrometry experiments, HEK293T cells were transfected with Flag‐YAP1, HA‐NSP13, or Flag‐YAP1 + HA‐NSP13 according to the manufacturer’s standard protocols. After nuclear extraction and lysis, the supernatant was incubated with HA magnetic beads to immunoprecipitate (IP) NSP13. The IP samples were subsequently analyzed by mass spectrometry to identify NSP13‐associated proteins (Fig. 4F). Each experimental condition was performed in duplicate to ensure reproducibility. We included an appropriate negative control (Flag‐YAP1) and stringent data‐filtering criteria to minimize false positives. We apologize for not including these details in our original Methods section; in this revised manuscript, we have fully described the number of replicates, the controls used, and our data analysis pipelines.